# Changes and prognostic value of cardiopulmonary exercise testing parameters in elderly patients undergoing cardiac rehabilitation: The EU-CaRE observational study

**Thimo Marcin**[1]\*, **Prisca Eser**[1], **Eva Prescott**[2], **Leonie F. Prins**[3], **Evelien Kolkman**[3], **Wendy Bruins**[4], **Astrid E. van der Velde**[4], **Carlos Peña Gil**[5], **Marie-Christine Iliou**[6], **Diego Ardissino**[7], **Uwe Zeymer**[8], **Esther P. Meindersma**[4,9], **Arnoud W. J. Van't Hof**[4,10,11], **Ed P. de Kluiver**[4], **Matthias Wilhelm**[1]

1 Department of Cardiology, Inselspital, Bern University Hospital, University of Bern, Bern, Switzerland, 2 Department of Cardiology, Bispebjerg Frederiksberg University Hospital, Copenhagen, Denmark, 3 Diagram B.V., Zwolle, The Netherlands, 4 Isala Heart Centre, Zwolle, The Netherlands, 5 Department of Cardiology, Complexo Hospitalario Universitario de Santiago de Compostela, SERGAS IDIS CIBERCV, Santiago, Spain, 6 Department of Cardiac Rehabilitation, Assistance Publique Hopitaux de Paris, Paris, France, 7 Department of Cardiology, Parma University Hospital, Parma, Italy, 8 Klinikum Ludwigshafen and Institut für Herzinfarktforschung Ludwigshafen, Ludwigshafen, Germany, 9 Department of Cardiology, Radboud University, Nijmegen, The Netherlands, 10 Department of Cardiology, Maastricht University Medical Center and Cardiovascular Research Institute Maastricht (CARIM), Maastricht, The Netherlands, 11 Department of Cardiology, Zuyderland Medical Center, Heerlen, The Netherlands

\* thimo.marcin@insel.ch

## Abstract

### Objective

We aimed 1) to test the applicability of the previously suggested prognostic value of CPET to elderly cardiac rehabilitation patients and 2) to explore the underlying mechanism of the greater improvement in exercise capacity (peak oxygen consumption, $VO_2$) after CR in surgical compared to non-surgical cardiac patients.

### Methods

Elderly patients ($\geq$65 years) commencing CR after coronary artery bypass grafting, surgical valve replacement (surgery-group), percutaneous coronary intervention, percutaneous valve replacement or without revascularisation (non-surgery group) were included in the prospective multi-center EU-CaRE study. CPETs were performed at start of CR, end of CR and 1-year-follow-up. Logistic models and receiver operating characteristics were used to determine prognostic values of CPET parameters for major adverse cardiac events (MACE). Linear models were performed for change in peak $VO_2$ (start to follow-up) and parameters accounting for the difference between surgery and non-surgery patients were sought.

**Data Availability Statement:** There are restrictions to data sharing but these are based on the various legislative restrictions for data containing

potentially identifying or sensitive patient information of the involved eight countries and the lack of informed consent to international data sharing. Requests for de-identified aggregated data may be sent to info@diagram-zwolle.nl, Diagram B. V., contract research organization, Dokter Stolteweg 96, 8025 AZ Zwolle, The Netherlands, on behalf of Prof. Arnoud Van't Hoff, Department of Cardiology, Maastricht University Medical Center, and Cardiovascular Research Institute Maastricht (CARIM), the Netherlands.

**Funding:** For the Swiss consortium partner (TM, PE, MW), funding was received by the Swiss State Secretariat for Education, Research and Innovation under contract number 15.0139. All other authors received funding by the European Union's Horizon 2020 research and innovation programme under grant agreement No 634439. The funder provided support in the form of salaries for authors [TM, PE, LP, EK, WB, AvV], but had no role in study design, data collection and analysis, decision to publish, or preparation of the manuscript. None of the commercial partners who provided financial support to any of the coauthors had any role in the perception, conduction, analysis, interpretation nor dissemination of this study.

**Competing interests:** AWJVH reports grants from Medtronic, grants and personal fees from Astra Zeneca, outside the submitted work, UZ reports grants and personal fees from Astra Zeneca, grants and personal fees from Bayer, per-sonal fees from Boehringer Ingelheim, grants and personal fees from BMS, personal fees from Daiichi Sankyo, personal fees from Eli Lilly, grants and personal fees from Novartis, grants and personal fees from MSD, personal fees from Trommsdorf, personal fees from Amgen, outside the submitted work. LP and EK work for Diagram B.V., a contract research organization. These commercial affiliations do not alter our adherence to all PLOS ONE policies on sharing data and materials. All other authors have no conflict of interest to declare.

**Abbreviations:** AUC, Area Under the Curve; BF, Breathing frequency; CABG, Coronary artery bypass graft; CPET, Cardiopulmonary exercise testing; CR, Cardiac rehabilitation; EU-CaRE, A European study on effectiveness and sustainability of current Cardiac Rehabilitation programmes in the Elderly; Hb, Haemoglobin; HR, Heart rate; MACE, Major Adverse Cardiac Event; OUES, Oxygen uptake efficiency slope; PCI, Percutaneous Coronary Intervention; RER, Respiratory Exchange Ratio; ROC, Receiver Operating Curves; TV, Tidal volume; $VCO_2$, Carbon dioxid output; VE, Ventilation; $VO_2$, Peak oxygen consumption; VT, Ventilatory threshold; Δ, Changes.

## Results

1421 out of 1633 EU-CaRE patients performed a valid CPET at start of CR (age 73±5.4, 81% male). No CPET parameter further improved the receiver operation characteristics significantly beyond the model with only clinical parameters. The higher improvement in peak $VO_2$ (25% vs. 7%) in the surgical group disappeared when adjusted for changes in peak tidal volume and haemoglobin.

## Conclusion

CPET did not improve the prediction of MACE in elderly CR patients. The higher improvement of exercise capacity in surgery patients was mainly driven by restoration of haemoglobin levels and improvement in respiratory function after sternotomy.

## Trial registration

Netherlands Trial Register, Trial NL5166.

## Introduction

Improving physical fitness is a cornerstone of modern cardiac rehabilitation (CR) [1] and a lack of improvement is associated with worse outcome [2–6]. The American Heart Association has recently emphasized functional physical capacity as a principal endpoint for therapies oriented to older adults with cardiovascular disease [7]. There is evidence that elderly CR patients are able to improve their physical fitness with CR, although the improvement seems to be attenuated with increasing age [8–10]. We previously reported that elderly cardiac patients after surgery have a lower physical fitness than patients with only minimal or no invasive procedure when commencing CR [11] and that they recover to the same level over the time course of one year [12]. A higher improvement in patients after coronary artery bypass graft (CABG) has also been shown in previous studies [13, 14], however, the underlying mechanisms of the recovery process has not been fully investigated to date. Peak oxygen consumption ($VO_2$) measured by cardiopulmonary exercise testing (CPET) is the gold standard for measuring physical fitness. Additionally, CPET provides a tool to characterise exercise limitation and differentiate between respiratory and circulatory patterns [15].

Besides peak $VO_2$, CPET provides additional parameters with prognostic value, namely the oxygen uptake efficiency slope (OUES), ventilation to carbon dioxide (VE/$CO_2$) slope, $VO_2$/ workload slope and the ventilatory thresholds ($VT_1$ and $VT_2$) [16]. Combining CPET parameters to a risk score has been shown to improve the prediction of adverse events in heart failure patients and coronary artery disease patients [16–18], but the predictive value for major cardiovascular adverse events (MACE) in elderly CR patients is unclear.

The study aims were 1) to determine prognostic values of CPET parameters for MACE after 1-year follow up in elderly patients commencing CR and 2) to identify respiratory and circulatory factors explaining the greater peak $VO_2$ improvement in surgical compared non-surgical patients from start of CR to 1-year follow-up.

## Materials and methods

The European Cardiac Rehabilitation in Elderly (EU-CaRE) study was a prospective cohort study performed from 2016 to 2019, with the aim to assesses the (cost-)effectiveness, sustainability and participation levels in current CR programs of eight cardiac rehabilitation centres in seven European countries (Denmark, France, Germany, the Netherlands, Italy, Spain and Switzerland) [19].

The study was approved by the lead ethics committee (Medisch Ethische Toetsingscommissie at Isala, Netherlands) and all relevant medical ethics committees of all participating centres:

Landesärztekammer Rheinland Pfalz, Germany (Nr. 837.341.15, (10109))

Comission Nationale de l'Informatique et de Libertés, France (DR-2016-021)

Secretario do Comité de Ética da Investigación de Santiago-Lugo, Spain (2015/486)

Comitato Ethico per Parma, Italy (34360)

Videnskabsetiske Komite C for Region Hovedstaden, Denmark (593)

Kantonale Ethikkomission Bern, Switzerland (290/15).

The study was registered at trialregister.nl (NTR5306). The participants gave written informed consent before they were included in the study.

### Study population

Patients with an age of $\geq$65 commencing CR after coronary artery bypass grafting, surgical valve replacement (surgery-group), percutaneous coronary intervention, percutaneous valve replacement or without revascularisation (non-surgery group) were consecutively included from January 2016 –January 2018.

Patients with a contraindication to CR, mental impairment leading to inability to cooperate, severe impaired ability to exercise, signs of severe cardiac ischemia and/or a positive exercise testing on severe cardiac ischemia, insufficient knowledge of the native language and an implanted cardiac device were excluded.

### Data collection and processing

Demographic, socioeconomic and cardiovascular risk factors as well as comorbidities were recorded through hospital records, interviewing, questionnaires and clinical assessments. Clinical assessments included CPET, anthropometric measurements, spirometry and resting heart rate. Haemoglobin was recorded if it was routinely determined in the clinical work up.

CPETs were performed on a cycle with an individualized ramp protocol aiming to achieve voluntary exhaustion within 8 to 12 min of ramp duration. CPET parameters were determined at the core lab in Bern by an automated procedure on raw data files using MATLAB (vers. R2017b, MathWorks[®], United States), as describe previously [11, 20]. One experienced operator (TM) performed extensive visual quality control using Wassermann's 9-panel plot and in case of doubtful quality, a second operator (MW) was consulted. Data from the gas exchange measurements were excluded from the analysis in case of suspected mask leakage or equipment failure, as well as if the ramp duration was less than 3 min.

Peak values from the CPET were determined as the highest value of a 30 s moving average and included peak $VO_2$, VE, breathing frequency (BF), tidal volume (TV) and oxygen pulse. The following submaximal gas measures were determined: VE/$VCO_2$ slope, $VO_2$/workload slope, the OUES, which represents the ratio of the log VE to $VO_2$. All ventilatory thresholds

($VT_1$ and $VT_2$) were visually determined by one single investigator (TM). Interrater reliability was determined in a random subset of 200 CPETs, in which thresholds were determined by a second experienced investigator (MW) blinded also to patients and centres as well as to thresholds set by the other investigator [20]. The respiratory exchange ratio (RER) as measure for exertion was determined by dividing $VCO_2$ by $VO_2$. Besides gas measures, further exercise parameters such as maximal workload, peak heart rate (HR), HR reserve (difference between peak and resting HR) and HR recovery after 60 s were recorded.

Adverse events, which were the primary outcome for this sub-study, were recorded by monthly telephone calls and assessed individually by an independent Clinical Event Committee. Major Adverse Cardiac Event (MACE) were defined as composite endpoint of all-cause and cardiovascular mortality, acute coronary syndrome, aborted sudden cardiac death and cardiovascular intervention/surgery, hospital admission or emergency visits between T0 and T2.

## Statistical analyses

All statistics were performed with R (Version 3.5.1, R Core Team, 2017).

Mixed logistic regression models (*lme4* package) adjusted for age, sex, PCI, time between index event and start of CR as fixed, centre as random intercept and baseline CPET parameters added individually to the model were performed to determine the associations of CPET characteristics with MACE. Existing cut-off values (peak $VO_2$ <18ml/kg/min, OUES <1550, VE/$CO_2$ slope >31.5) were used to compare the risk of MACE between patients with and without impaired CPET characteristics at start of CR [16]. Additionally, optimal specific cut-offs for peak $VO_2$, OUES and VE/$CO_2$ slope were determined for our surgery and non-surgery patients using receiver operator characteristics (ROC) and Youden's index with 95% confidence intervals (CI) calculated by bootstrapping (*Cutpointr* package). We compared the area under the curve (AUC) of each model including the CPET parameter in question to the model without any CPET parameters using bootstrap test for two ROC curves (*pROC* package).

CPET characteristics were compared between surgery and non-surgery patients using t-tests for T0 and T2. Improvement in percent and Cohen's D effect sizes were calculated to compare changes (Δ) between the CPET parameters. Additionally, linear models robust for outliers (package *robustbase*) were performed to explore whether the difference in Δpeak $VO_2$ between surgery and non-surgery patients may be explained by respiratory (ΔTV, ΔBF) or circulatory/peripheral changes (ΔHR reserve, Δhaemoglobin, $ΔVT_1$). We performed available case analyses.

Alpha was set at 0.01 for all analyses instead of 0.05 to adjust somewhat for multiple testing. Residual plots were used to check model assumptions (normality, variance homogeneity and linearity) in the linear robust models and deviance statistics assessed for the logistic models.

## Results and discussion

1421 out of 1633 EU-CaRE patients performed a CPET with acceptable test quality before start of CR and 1178 as well at one-year follow-up (Fig 1).

Main characteristics of the 1178 patients are presented in Table 1. The characteristics of the comparable full EU-CaRE population has been reported elsewhere [20–23]. From the 1421 patients, 195 (14%) reported a MACE within a mean (SD) follow up time of 340 (112) days, namely 14 (1%) all-cause-mortality, 11 (1%) CV-mortality, 1 (0%) aborted sudden cardiac death, 26 (2%) acute coronary syndrome, 121 (9%) CV hospitalisations, 107 (8%) CV emergency visits and 123 (9%) CV interventions.

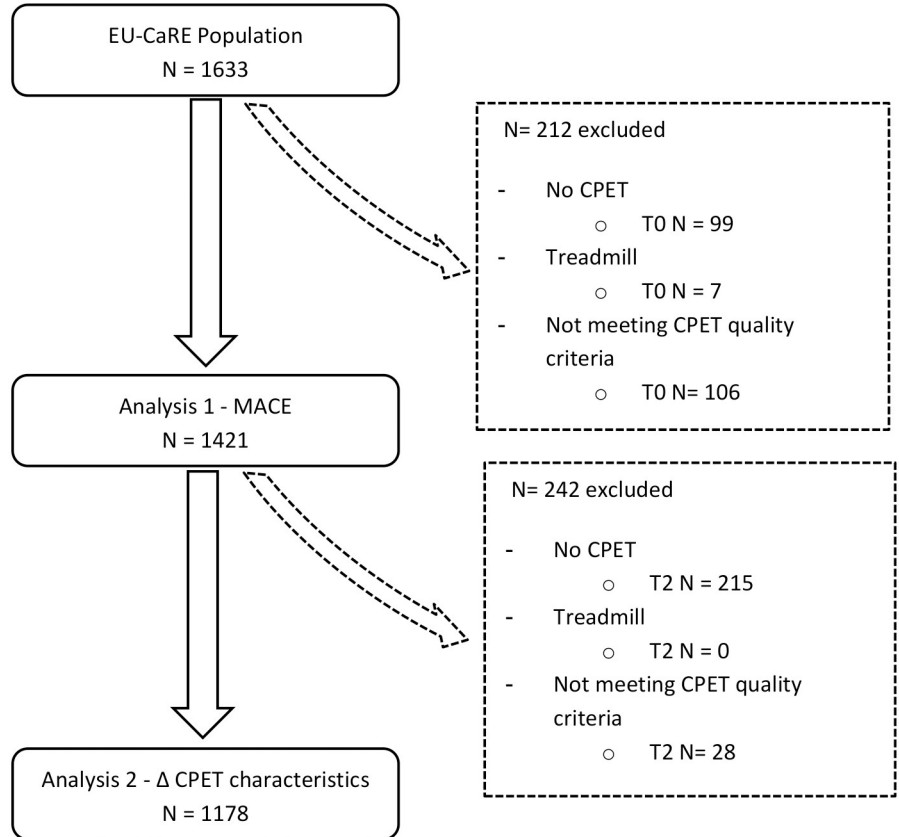

**Fig 1. Flow-chart of the patients included for the analyses of MACE and changes in CPET characteristics.** T0, start of CR; T2, 1-year follow-up.

Higher peak VO$_2$, OUES, VE/CO$_2$-slope at baseline was associated with lower risk for MACE in one-year follow-up, when adjusted for age, sex, PCI and time from index to start of CR (Table 2).

Patients with impaired values in all three variables had the highest risk of MACE. The cut-offs with 95% CI derived from our own study population for the non-surgery and surgery group were as follow: peak VO$_2$, 15.7 [11.8–18.1] ml/kg/min and 12.5 [9.8–15.7]; OUES, 1.75 [1.2–2.1] and 1.35 [0.58–2.26]; VE/CO$_2$-slope, 50.1 [27.4–58.6] and 34.2 [31.5–38.2]. Using our own cut-offs did not significantly improve the prediction of MACE (AUC = 66.99, specificity = 52.86, sensitivity = 73.84) compared to the established cut-offs (Table 2) [16]. Additionally, impaired oxygen pulse and VE/CO$_2$-slope standardised for peak VO$_2$ were associated with an increased risk of MACE. Overall, no single CPET parameter significantly improve the AUC compared to the multivariate logistic model without the respective CPET parameter. From the potentially confounding factors included in the model, only PCI as indication for CR was associated with MACE (Odds ratio ≈ 1.7)x, probably driven by the great proportion of patients with PCI after an acute coronary syndrome. Age, sex or time lag of CR uptake did not significantly predict cardiac events (full output shown in S1 Table). Analysis of deviance did not indicate a lack of fit in any of the performed logistic models.

S1 Fig in the supplemental information shows the survival curves for MACE in patients after surgery and non-surgery and patients with the CPET risk score 1–4 based on our own cut-offs, illustrating the distribution of MACE over time.

**Table 1. Baseline characteristics.**

| Variable | All | Surgery | Non-surgery |
|---|---|---|---|
| | N = 1178 | n = 423 | n = 755 |
| Age [y] | 72.5 (5.3) | 72.6 (4.9) | 72.41 (5.49) |
| Male Sex | 957 (81%) | 372 (88%) | 581 (77%) |
| Ejection Fraction [%] | | | |
| >55 | 614 (58%) | 241 (64%) | 373 (54%) |
| 45–55 | 291 (27%) | 98 (26%) | 193 (28%) |
| 35–44 | 123 (12%) | 30 (8%) | 93 (13.5%) |
| <35 | 36 (3%) | 6 (1.6%) | 30 (4.3%) |
| Acute Coronary Syndrome | 654 (56%) | 80 (19%) | 573 (76%) |
| Procedure | | | |
| PCI | 653 (55%) | | |
| Chronic CAD without revascularization | 78 (7%) | | |
| Percutaneous valve replacement | 101 (2%) | | |
| Surgical valve replacement | 79 (7%) | | |
| CABG | 344 (29%) | | |
| Diabetes mellitus | 270 (23%) | 96 (23%) | 174 (23%) |
| COPD | 68 (6%) | 21 (5%) | 47 (6%) |

Values are meand (SD) and counts (percentage) as appropriate. SD, standard deviation; PCI, percutaneous coronary intervention; CABG, coronary artery bypass grafting; COPD, chronic obstructive pulmonary disease.

Fig 2 shows the comparison of the CPET characteristics (including resting HR and haemo-globin as additional CPET related parameters) between the surgery and non-surgery patients for T0 and T2 and the changes between the two time-points. At start of CR, most CPET parameters differed significantly between the two groups. In contrast, there were no significant

**Table 2. Multiple logistic mixed models for major adverse cardiac events.**

| CPET Predictors[a] | | OR | 99% CI | AUC [%] | Specifity [%][b] | Sensitivity [%] [b] | p-value[c] |
|---|---|---|---|---|---|---|---|
| Peak VO$_2$ [per SD] | | 0.73 | (0.57; 0.93)* | 64.61 | 49.10 | 75.90 | 0.08 |
| VE/VCO$_2$-Slope [per SD] | | 1.23 | (1.01; 1.52)* | 63.49 | 61.04 | 63.07 | 0.62 |
| OUES [per SD] | | 0.75 | (0.59; 0.95)* | 63.86 | 65.58 | 57.44 | 0.53 |
| VE/VCO$_2$slope/VO$_2$ [per SD] | | 1.31 | (1.07; 1.60)* | 64.90 | 48.77 | 76.97 | 0.25 |
| VT1 [per SD] | | 0.90 | (0.70; 1.19) | 65.53 | 45.63 | 76.17 | 0.58 |
| O$_2$-pulse [per SD] | | 0.75 | (0.59; 0.96)* | 63.63 | 69.37 | 52.81 | 0.65 |
| HR recovery [per SD] | | 0.84 | (0.66; 1.07) | 63.36 | 52.74 | 69.40 | 0.64 |
| HR reserve [per SD] | | 0.90 | (0.70; 1.13) | 63.39 | 68.90 | 55.38 | 0.32 |
| CPET RISK SCORE[c] (reference 0) | 1 | 1.90 | (0.89; 4.06) | 64.89 | 57.53 | 67.69 | 0.33 |
| | 2 | 2.20 | (1.02; 4.76)* | | | | |
| | 3 | 3.06 | (1.42; 6.62)* | | | | |

[a] adjusted for age, sex, timelag of CR uptake, PCI as fixed, centre as random factor and the respective CPET parameter.

[b] Best classification threshold according to Youden-Index.

[c] bootstrap test for two correlated R curves (model with vs. model without CPET parameter).

[d] Number of values below cut-off in peak VO$_2$ <18ml/kg/min, OUES <1.55 and VE/CO$_2$ slope >31.5 [16].

CPET, cardiopulmonary exercise testing; OR, Odds Ratio; CI, Confidence Interval; VO$_2$, oxygen consumption; VE, ventilation; OUES, oxygen uptake efficency slope; VCO$_2$, carbon dioxid output; HR, heart rate.

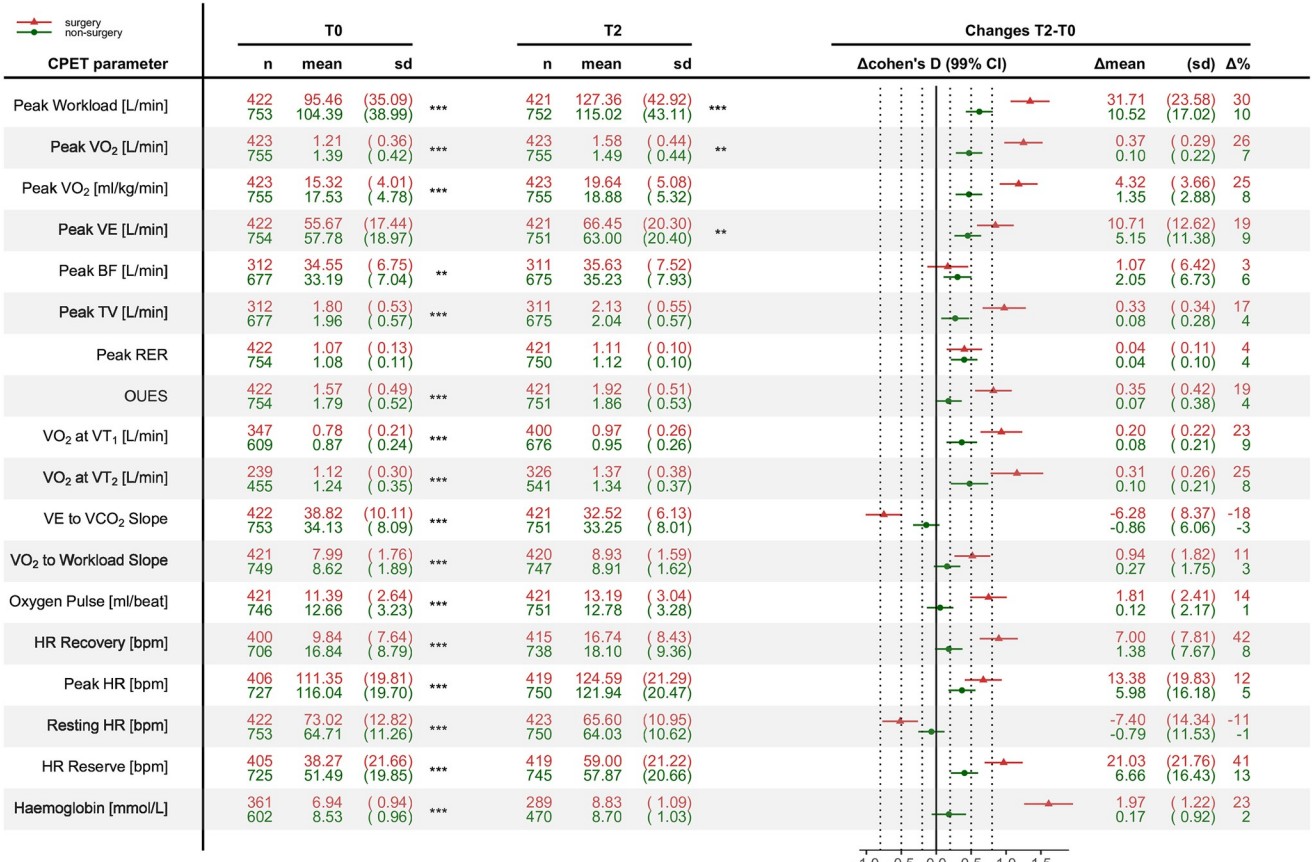

| CPET parameter | T0 n | mean | sd | | T2 n | mean | sd | | Changes T2-T0 Δcohen's D (99% CI) | Δmean | (sd) | Δ% |
|---|---|---|---|---|---|---|---|---|---|---|---|---|
| Peak Workload [L/min] | 422 / 753 | 95.46 / 104.39 | (35.09) / (38.99) | *** | 421 / 752 | 127.36 / 115.02 | (42.92) / (43.11) | *** | | 31.71 / 10.52 | (23.58) / (17.02) | 30 / 10 |
| Peak VO$_2$ [L/min] | 423 / 755 | 1.21 / 1.39 | (0.36) / (0.42) | *** | 423 / 755 | 1.58 / 1.49 | (0.44) / (0.44) | ** | | 0.37 / 0.10 | (0.29) / (0.22) | 26 / 7 |
| Peak VO$_2$ [ml/kg/min] | 423 / 755 | 15.32 / 17.53 | (4.01) / (4.78) | *** | 423 / 755 | 19.64 / 18.88 | (5.08) / (5.32) | | | 4.32 / 1.35 | (3.66) / (2.88) | 25 / 8 |
| Peak VE [L/min] | 422 / 754 | 55.67 / 57.78 | (17.44) / (18.97) | | 421 / 751 | 66.45 / 63.00 | (20.30) / (20.40) | ** | | 10.71 / 5.15 | (12.62) / (11.38) | 19 / 9 |
| Peak BF [L/min] | 312 / 677 | 34.55 / 33.19 | (6.75) / (7.04) | ** | 311 / 675 | 35.63 / 35.23 | (7.52) / (7.93) | | | 1.07 / 2.05 | (6.42) / (6.73) | 3 / 6 |
| Peak TV [L/min] | 312 / 677 | 1.80 / 1.96 | (0.53) / (0.57) | *** | 311 / 675 | 2.13 / 2.04 | (0.55) / (0.57) | | | 0.33 / 0.08 | (0.34) / (0.28) | 17 / 4 |
| Peak RER | 422 / 754 | 1.07 / 1.08 | (0.13) / (0.11) | | 421 / 750 | 1.11 / 1.12 | (0.10) / (0.10) | | | 0.04 / 0.04 | (0.11) / (0.10) | 4 / 4 |
| OUES | 422 / 754 | 1.57 / 1.79 | (0.49) / (0.52) | *** | 421 / 751 | 1.92 / 1.86 | (0.51) / (0.53) | | | 0.35 / 0.07 | (0.42) / (0.38) | 19 / 4 |
| VO$_2$ at VT$_1$ [L/min] | 347 / 609 | 0.78 / 0.87 | (0.21) / (0.24) | *** | 400 / 676 | 0.97 / 0.95 | (0.26) / (0.26) | | | 0.20 / 0.08 | (0.22) / (0.21) | 23 / 9 |
| VO$_2$ at VT$_2$ [L/min] | 239 / 455 | 1.12 / 1.24 | (0.30) / (0.35) | *** | 326 / 541 | 1.37 / 1.34 | (0.38) / (0.37) | | | 0.31 / 0.10 | (0.26) / (0.21) | 25 / 8 |
| VE to VCO$_2$ Slope | 422 / 753 | 38.82 / 34.13 | (10.11) / (8.09) | *** | 421 / 751 | 32.52 / 33.25 | (6.13) / (8.01) | | | -6.28 / -0.86 | (8.37) / (6.06) | -18 / -3 |
| VO$_2$ to Workload Slope | 421 / 749 | 7.99 / 8.62 | (1.76) / (1.89) | *** | 420 / 747 | 8.93 / 8.91 | (1.59) / (1.62) | | | 0.94 / 0.27 | (1.82) / (1.75) | 11 / 3 |
| Oxygen Pulse [ml/beat] | 421 / 746 | 11.39 / 12.66 | (2.64) / (3.23) | *** | 421 / 751 | 13.19 / 12.78 | (3.04) / (3.28) | | | 1.81 / 0.12 | (2.41) / (2.17) | 14 / 1 |
| HR Recovery [bpm] | 400 / 706 | 9.84 / 16.84 | (7.64) / (8.79) | *** | 415 / 738 | 16.74 / 18.10 | (8.43) / (9.36) | | | 7.00 / 1.38 | (7.81) / (7.67) | 42 / 8 |
| Peak HR [bpm] | 406 / 727 | 111.35 / 116.04 | (19.81) / (19.70) | *** | 419 / 750 | 124.59 / 121.94 | (21.29) / (20.47) | | | 13.38 / 5.98 | (19.83) / (16.18) | 12 / 5 |
| Resting HR [bpm] | 422 / 753 | 73.02 / 64.71 | (12.82) / (11.26) | *** | 423 / 750 | 65.60 / 64.03 | (10.95) / (10.62) | | | -7.40 / -0.79 | (14.34) / (11.53) | -11 / -1 |
| HR Reserve [bpm] | 405 / 725 | 38.27 / 51.49 | (21.66) / (19.85) | *** | 419 / 745 | 59.00 / 57.87 | (21.22) / (20.66) | | | 21.03 / 6.66 | (21.76) / (16.43) | 41 / 13 |
| Haemoglobin [mmol/L] | 361 / 602 | 6.94 / 8.53 | (0.94) / (0.96) | *** | 289 / 470 | 8.83 / 8.70 | (1.09) / (1.03) | | | 1.97 / 0.17 | (1.22) / (0.92) | 23 / 2 |

(surgery / non-surgery; x-axis: -1.0 -0.5 0.0 0.5 1.0 1.5)

**Fig 2. CPET characteristics in surgery and non-surgery patients.** Shown are mean and standard deviation of all CPET and exercise related parameters at start of CR (T0) and 1 year follow-up (T2) as well the changes as standardised effect size (cohen's D). Cohen's Ds of 0.2 indicate a weak, 0.4 a medium and 0.8 a large effect size. VO2, oxygen consumption; VE, ventilation; BF, breathing frequency; TV, tidal volume; RER, respiratory exchange ratio; OUES, oxygen uptake efficiency slope; VT, ventilatory threshold; HR, heart rate.

differences at 1-year follow-up, except for peak Watt, absolute peak VO2 [L/min] and peak VE [L/min].

Mean changes (T2-T0) of the CPET characteristics are illustrated as Cohen's D effect size (mean/standard deviation) with 99% confidence interval (CI) in Fig 2. Changes in submaximal parameters, namely the OUES, VE/CO$_2$-slope or the VT$_1$ were only slightly lower than changes in peak exercise variables such as peak VO$_2$ and peak Workload. HR reserve and HR recovery improved most with 41 and 42% in surgery patients and 13 and 8% in non-surgery whereas the effect size was largest in haemoglobin (ΔHb) due to the relatively low standard deviation.

Mean improvement in peak VO$_2$ was 0.25 l/min higher in surgery patients compared to non-surgery patients. However, the difference declined when adjusting for ΔHb, ΔVT1 or ΔHR reserve, was more than halved when adjusted for Δpeak TV and disappeared almost completely when adjusted for ΔHb and Δpeak TV variables (Fig 3). Adding change in RER to the model in order to account for the potential confounding effect of submaximal CPETs did not influence the results. Model diagnostic did not indicate violation of model assumptions in any of the performed robust linear model.

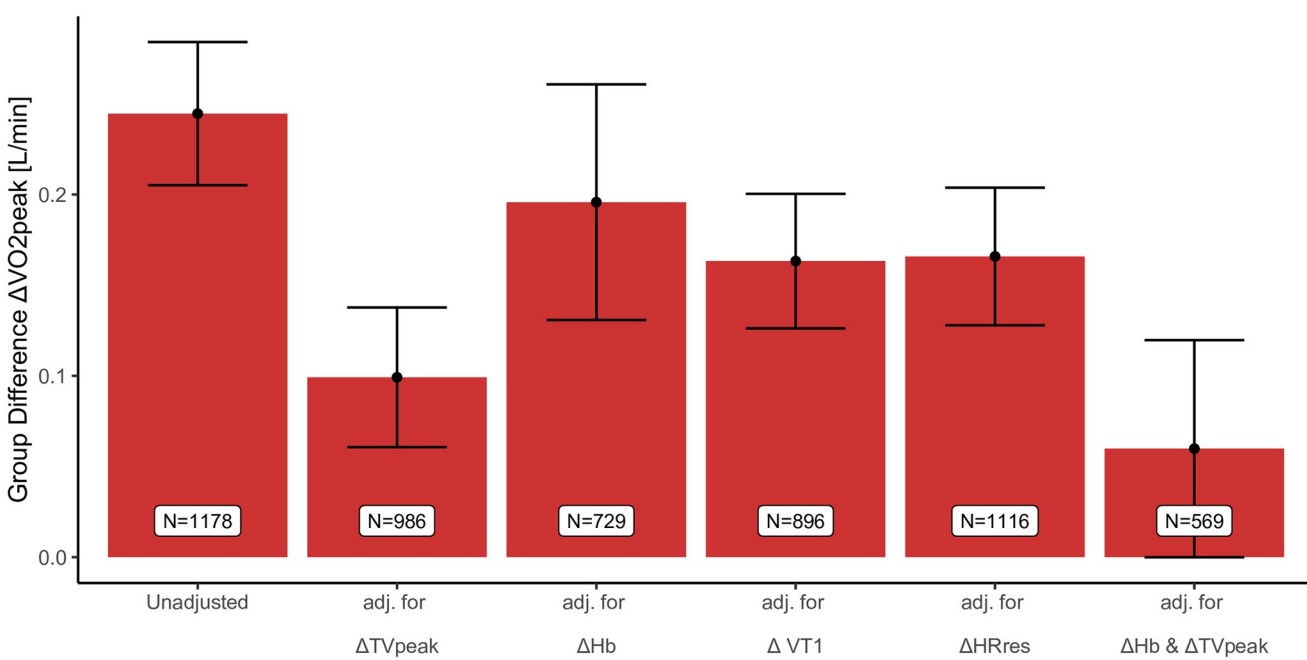

**Fig 3. Mean differences (99%CI) in Δpeak VO2 [ml/kg/min] between surgery and non-surgery patients when adjusted for respiratory and circulatory/peripheral CPET parameters.** adj., adjusted; VO2, oxygen consumption; TV, tidal volume; Hb, haemoglobin; VT, ventilatory threshold; HRres, heart rate reserve.

## Discussion

Patients with an impaired VE/CO$_2$slope, peak VO$_2$ and OUES and oxygen pulse at start of CR were at higher risk of developing a major cardiac adverse event within one year. However, no single CPET parameter significantly improved the prediction of a multivariate logistic model including presence of PCI and age. Nevertheless, the present study provides a detailed insight into the CPET characteristics of 1178 elderly cardiac patients participating in current European CR programmes. Our data suggests that the greater improvement in peak VO$_2$ in surgery compared to non-surgery patients was mainly driven by changes in peak TV and Hb, and based on their lower pre-CR values.

### Major adverse cardiac events

In a previous study, combining CPET parameters has been found to add prognostic information, namely patients with an OUES<1550, a VE/VCO$_2$ slope >31.5 and VO$_2$ peak <18.3 ml/kg/min were more likely to develop MACE compared to patients with a normal values or bad performance in only one or two of these variables [16]. Similarly in our study, patients with a value below these cut-offs showed a 3.14 fold risk of MACE in the one-year follow-up compared to patients with values above the cut-offs. However, the patients of the present study were older and most likely weaker than in the study from Coeckelberghs et al [16]. Hence, the cut-offs were probably not appropriate and the predictive value correspondingly underestimated. We therefore calculated cut-off values based on our own elderly cohort, nonetheless, these cut-offs did not significantly improve the discriminative performance compared to the established cut-offs. The 95% CI of our own cut-offs were wide and may therefore not be applicable for other cohorts. Overall, the models were poor in predicting MACE indicated by a very low AUC (<65). Guazzi et al. found an improved prediction of survival in chronic heart failure

patients when the VE/$CO_2$-slope was normalised for peak $VO_2$ [24], however, neither this index nor any other CPET parameter significantly improved the discriminative performance for MACE in our elderly CR patients. In our study, the follow-up period may have been too short and the definition of MACE too wide to obtain a valuable prediction of MACE.

## Changes in CPET characteristics

Patients after open chest surgery, namely CABG and surgical valve replacement, are generally more deconditioned at start of CR than patients after percutaneous intervention or without revascularization as reported elsewhere [25]. This is also reflected by the overall deteriorated CPET characteristics as shown in the present study (Fig 2). Maximal exercise parameters were significantly reduced in surgery patients despite similar level of exertion (peak RER). Submaximal parameters related to exercise capacity ($VT_1$, $VT_2$) and ventilatory efficiency (OUES, VE/$CO_2$slope) were also reduced. Surgery patients showed overall a larger improvement in the CPET characteristics (medium to large effect size) and differences to non-surgery were mostly abolished at one-year follow-up (Fig 2). Similar findings were reported by Lan et. al who observed lower baseline values and greater improvements of peak $VO_2$ and $VT_1$ in CABG patients compared to PCI patients [26].

There is likely a greater spontaneous recovery in surgery patients than non-surgery patients. This recovery process may be enhanced by CR, however the evidence is weak and the beneficial effect may only account for patients with reduced ventricular function [27]. A randomized trial found similar improvements in peak $VO_2$ in the CR group and the control group [28].

As shown in Fig 2, the higher improvement in peak $VO_2$ in surgery patients could be explained by the larger improvement in peak TV and Hb whereas chronotropic changes (HR reserve) contributed only little to the differences in peak $VO_2$. It has been shown that patients after sternotomy suffer from an impaired lung volume capacity and reduced respiratory muscle strength 6 days postoperative [29], but recover their respiratory muscle function 2 months after surgery [30]. Exercise training has shown to improve ventilatory pattern by improving the rapidness and depth of breathing during exercise in patients with heart disease [31]. Similarly, postoperative inspiratory muscle training in patients undergoing cardiac surgery has been found to improve maximal inspiratory pressure, tidal volume and peak expiratory flow [29]. Inspiratory muscle training may therefore be used in the CR of cardiac surgery patients in order to improve their exercise capacity, but also in elderly fragile non-surgery patients unable to exercise.

A recent study assigned a contributing role of autonomic function to the peak $VO_2$ improvements in coronary artery disease patients undergoing CR [32]. They found an improvement in the chronotropic response in the responder group ($\Delta$ peak $VO_2$ >2.6ml/kg/min) but no improvements in the non-responder group ($\Delta$ peak $VO_2 \leq$ 2.6ml/kg/min). In this study, surgery patients improved their HR reserve much more than non-surgery patients (40% vs. 13%) but the larger improvement in HR reserve did not explain the larger improvement in peak $VO_2$. In contrast to respiratory function, improvement in chronotropic response seemed to have a lower impact on changes in exercise capacity.

In accordance to a prior study that found a significant association of $\Delta$Hb and improvements in peak $VO_2$ in CABG patients [33], we found that in surgery patients Hb largely recovered (Cohen's D > 1.5) within one year. Given these results, it is not surprising that $\Delta$Hb explained partly the higher improvement in peak $VO_2$ of the surgery patients.

Early onset of the anaerobic threshold (reflected by early $VT_1$) occurs in anaemic as well as patients with muscular deconditioning [34], and an improvement in the threshold may reflect circulatory and/or peripheral improvements. As $\Delta$Hb explained as much as $\Delta VT_1$ of the

difference in Δpeak $VO_2$ between these groups, it is likely that exercise capacity in elderly surgery patients improves via restoration of Hb levels and improved respiratory function, and less by circulatory or peripheral improvement.

## Strengths

This is a large multi-centre study of a commonly underrepresented elderly cardiac patient population. All CPET data have been automatically analysed in the Core Lab of Bern. Reporting the CPET characteristics as outcomes of CR allows a more comprehensive assessment of exercise performance and enables to discriminate between respiratory and circulatory/peripheral changes.

## Limitations

The present study is part of the EU-CaRE study that primarily aimed to compare the Δpeak $VO_2$ between the participating rehabilitation centres. Therefore, the presented analyses are of explorative nature and the associations cannot infer causality. Not all included patients performed a high quality CPET, and in a considerably large proportion (19%) of patients the $VT_1$ could not be determined. However, CPET duration was on average 7.75 min (SD 2.7) at baseline and 9.0 min (SD 2.8) at 1-year follow up and therefore of acceptable test duration. In addition, the effect of peak VO2 on MACE was not altered when the logistic model was adjusted for RER. Further, Hb was not routinely assessed in all centres and ΔHb was therefore missing in 62% of the included patients. However, the $\Delta VO_2$peak was comparable between patients with and without missing values. Nevertheless, patients without CPET data tended to have a higher risk for MACE (OR 1.58, p = 0.0432).

## Conclusion

CPET parameters did not add to the prediction of major adverse cardiovascular events within one year in this large elderly cohort. Submaximal as well as maximal CPET parameters improved significantly more in patients after open chest surgery compared to patients with no or minimally invasive intervention. The higher improvement of exercise capacity in elderly surgery patients was mainly driven by restoration of haemoglobin levels and improvement in respiratory function after sternotomy. In clinical studies on peak $VO_2$, the potentially large confounding effect of haemoglobin should be considered. Supportive respiratory muscle training may be beneficial in elderly cardiac surgery patients.

## Supporting information

**S1 Fig. Kaplan-Meier curves for major adverse cardiovascular events within 365 days after cardiac rehabilitation entry.** Panel A shows patients after cardiac surgery and no surgery. Panel B shows patients by CPET risk score (reduced peak VO2, VE to VCO2 slope and/or OUES based on the cut-offs derived from this study). VO2, oxygen uptake; VE, ventilation; VCO2, carbon dioxide ouput; OUES, oxygen uptake efficency slope.
(DOCX)

**S1 Table. Multiple logistic mixed models for major adverse cardiac events.**
(DOCX)

**S1 File.**
(PDF)

**S2 File.**
(PDF)

## Acknowledgments

We greatly appreciated the support by all not personally named people involved in the EU-CaRE study.

## Author Contributions

**Conceptualization:** Thimo Marcin, Prisca Eser, Eva Prescott, Wendy Bruins, Astrid E. van der Velde, Carlos Peña Gil, Marie-Christine Iliou, Diego Ardissino, Uwe Zeymer, Arnoud W. J. Van't Hof, Ed P. de Kluiver, Matthias Wilhelm.

**Data curation:** Thimo Marcin, Eva Prescott, Leonie F. Prins, Evelien Kolkman.

**Formal analysis:** Thimo Marcin, Prisca Eser.

**Funding acquisition:** Eva Prescott, Wendy Bruins, Astrid E. van der Velde, Carlos Peña Gil, Marie-Christine Iliou, Diego Ardissino, Uwe Zeymer, Arnoud W. J. Van't Hof, Ed P. de Kluiver, Matthias Wilhelm.

**Investigation:** Thimo Marcin, Prisca Eser, Leonie F. Prins, Wendy Bruins, Astrid E. van der Velde, Carlos Peña Gil, Marie-Christine Iliou, Diego Ardissino, Uwe Zeymer, Esther P. Meindersma, Arnoud W. J. Van't Hof, Ed P. de Kluiver, Matthias Wilhelm.

**Methodology:** Thimo Marcin, Prisca Eser, Evelien Kolkman, Astrid E. van der Velde, Carlos Peña Gil, Marie-Christine Iliou, Diego Ardissino, Uwe Zeymer, Esther P. Meindersma, Arnoud W. J. Van't Hof, Ed P. de Kluiver, Matthias Wilhelm.

**Project administration:** Thimo Marcin, Eva Prescott, Leonie F. Prins, Evelien Kolkman, Wendy Bruins, Astrid E. van der Velde, Carlos Peña Gil, Marie-Christine Iliou, Diego Ardissino, Uwe Zeymer, Esther P. Meindersma, Arnoud W. J. Van't Hof, Ed P. de Kluiver, Matthias Wilhelm.

**Supervision:** Prisca Eser, Eva Prescott, Arnoud W. J. Van't Hof, Matthias Wilhelm.

**Validation:** Thimo Marcin, Prisca Eser, Leonie F. Prins, Evelien Kolkman.

**Visualization:** Thimo Marcin.

**Writing – original draft:** Thimo Marcin.

**Writing – review & editing:** Thimo Marcin, Prisca Eser, Eva Prescott, Leonie F. Prins, Evelien Kolkman, Wendy Bruins, Astrid E. van der Velde, Carlos Peña Gil, Marie-Christine Iliou, Diego Ardissino, Uwe Zeymer, Esther P. Meindersma, Arnoud W. J. Van't Hof, Ed P. de Kluiver, Matthias Wilhelm.

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
