## [Decision Letter · Decision Letter 0]

11 Feb 2021

PONE-D-20-37659

Changes and prognostic value of cardiopulmonary exercise testing parameters in elderly patients undergoing cardiac rehabilitation: the EU-CaRE observational study

PLOS ONE

Dear Dr. Marcin,

Thank you for submitting your manuscript to PLOS ONE. After careful consideration, we feel that it has merit but does not fully meet PLOS ONE’s publication criteria as it currently stands. Therefore, we invite you to submit a revised version of the manuscript that addresses the points raised during the review process.

We look forward to receiving your revised manuscript.

Kind regards,

Gerson Cipriano Jr., PT, MsC, Ph.D.

Academic Editor

PLOS ONE

Additional Editor Comments:

After careful consideration, we feel that it has merit but does not fully meet PLOS ONE's publication criteria as it currently stands.

Therefore, we invite you to submit a revised version of the manuscript that addresses the concerns raised by reviewers;

their comments are available below.

Journal Requirements:

2. Thank you for including your ethics statement: "The study was approved by all relevant medical ethics committees of all participating centres, and registered at trialregister.nl (NTR5306)."

"For the Swiss consortium partner, funding was received by the

Swiss State Secretariat for Education, Research and Innovation under contract number 15.0139."

"The study was funded by the European Union’s Horizon 2020 research and innovation program under grant agreement number 634439 (Arnoud Van't Hoff) and by the Swiss State Secretariat for Education, Research and Innovation for the Swiss consortium partner (Matthias Wilhelm)."

"I have read the journal's policy and the authors of this manuscript have the following competing interests: AWJvH reports grants from Medtronic, grants and personal fees from Astra Zeneca, outside the submitted work, UZ reports grants and personal fees from Astra Zeneca, grants and personal fees from Bayer, personal fees from Boehringer Ingelheim, grants and personal fees from BMS, personal fees from Daiichi Sankyo, personal fees from Eli Lilly, grants and personal fees from Novartis, grants and personal fees from MSD, personal fees from Trommsdorf, personal fees from Amgen, outside the submitted work. The other authors have no conflict of interest to declare."

We note that one or more of the authors are employed by a commercial company: Diagram B.V.

5.1. Please provide an amended Funding Statement declaring this commercial affiliation, as well as a statement regarding the Role of Funders in your study. If the funding organization did not play a role in the study design, data collection and analysis, decision to publish, or preparation of the manuscript and only provided financial support in the form of authors' salaries and/or research materials, please review your statements relating to the author contributions, and ensure you have specifically and accurately indicated the role(s) that these authors had in your study. You can update author roles in the Author Contributions section of the online submission form.

5.2. Please also provide an updated Competing Interests Statement declaring this commercial affiliation along with any other relevant declarations relating to employment, consultancy, patents, products in development, or marketed products, etc.  

6.  We note that you have indicated that data from this study are available upon request. PLOS only allows data to be available upon request if there are legal or ethical restrictions on sharing data publicly. For information on unacceptable data access restrictions, please see http://journals.plos.org/plosone/s/data-availability#loc-unacceptable-data-access-restrictions.

Reviewers' comments:

Reviewer's Responses to Questions

**Comments to the Author**

1. Is the manuscript technically sound, and do the data support the conclusions?

Reviewer #1: Yes

Reviewer #2: Partly

2. Has the statistical analysis been performed appropriately and rigorously? 

Reviewer #1: Yes

Reviewer #2: No

3. Have the authors made all data underlying the findings in their manuscript fully available?

Reviewer #1: No

Reviewer #2: No

4. Is the manuscript presented in an intelligible fashion and written in standard English?

Reviewer #1: Yes

Reviewer #2: Yes

5. Review Comments to the Author

Reviewer #1: PONE-D-20-37659: statistical review

SUMMARY. This is a longitudinal study that (1) tests whether parameters of cardiopulmonary exercise testing (CPET) are significant predictors of major cardiovascular adverse events (MACE) in elderly patients who commence cardiac rehabilitation and (2) seeks factors that explain the greater peak VO2 improvement in surgical compared non-surgical patients. Overall, the statistical analysis seems appropriate, but the authors should describe additional details about the exploited methods (see major issues 1 and 2 below). Furthermore, model diagnostics should be provided (major issue 3).

MAJOR ISSUES

1. The statistical analysis that addresses aim (1) appropriately relies on a mixed effects logistic model. However, Table 2 does not display the parameters of the random effect distribution. I was therefore unable to understand the specific random effect correlation structure that has been chosen for this analysis. The authors should add details about this correlation structure and display the estimated parameters in Table 2.

2. The analysis of the factors that explain VO2 improvements is based on "robust linear models". However, it is not clear what kind of model has been used by the authors. After all, robustness has many aspects! Robustness with respect to outliers? Robustness with respect to variance assumptions? This should be clarified.

3. It seems (last paragraph of the statistical analysis) that traditional model diagnostics have been performed, which is a good idea. However, these results are not displayed nor commented. At least, the results of the diagnostics should be summarized in a supplementary file.

Reviewer #2: The study had two objectives:

1) to test the applicability of the previously suggested prognostic value of CPET to elderly cardiac rehabilitation (CR) patients.

2) to explore the underlying mechanism of the greater improvement in exercise capacity (peak oxygen consumption, VO2) after CR in surgical compared to

non-surgical cardiac patients.

Cardiopulmonary exercise tests (CPET) were performed at the start of CR, end of CR, and 1-year follow-up.

Patients were divided into two groups.

- Surgery group: after coronary artery bypass grafting, surgical valve replacement.

- Non-surgery group: percutaneous coronary intervention, percutaneous valve replacement, or without revascularization.

Adverse Cardiac Event (MACE) was defined as a composite endpoint of all-cause and cardiovascular mortality, acute coronary syndrome, aborted sudden cardiac death, cardiovascular intervention/surgery, hospital admission, or emergency visits between T0 and T2.

Results

No CPET parameter further improved the receiver operation characteristics significantly with the model, including only clinical parameters. The higher improvement in peak VO2 (25% vs. 7%) in the surgical group disappeared when adjusted for peak tidal volume and hemoglobin changes.

Conclusion

CPET did not improve the prediction of MACE in elderly CR patients. The more significant improvement of exercise capacity in surgery patients was mainly driven by the restoration of hemoglobin levels and improvement in respiratory function after sternotomy.

My considerations about the article:

It is an interesting paper about cardiovascular rehabilitation (CR) on elderly patients (age 73±5,4, 81% male), with a good number of participants (1,421 for prognostic evaluation and 1,178 for modification on CPET) and multicentric (eight centers in seven countries).

1) Although the CPET aimed a duration of 8-12 minutes, exclusion criteria excluded a protocol duration shorter than 3 minutes. Were CPET with a duration of 3 to 8 minutes included for analyses? These CPETs with short duration (mainly those with less than 5-6 minutes) may have limited the measured variables' utilization and, therefore, the prognostic value.

CPET duration (mean and SD) was not shown in Figure 2 or described in the article.

2) Existing cut-off values (peak VO2 <18 ml/kg/min, OUES <1550, VE/CO2 slope >31.5) were used to compare the risk of MACE between patients with and without impaired CPET characteristics at the start of CR. [Ref. 16]

[16] Coeckelberghs E, Buys R, Goetschalckx K, Cornelissen VA, Vanhees L. Prognostic value of the oxygen uptake efficiency slope and other exercise variables in patients with coronary artery disease. European journal of preventive cardiology 2016;23(3):237–44.

These cut-offs values were determined by a previous study with a younger population (60,7±9,9 years), all patients with CAD, and a higher peak VO2 (19,5±5,6 ml/kg/min), compared to the surgery group of the presented manuscript (15,3±4,0 ml/kg/min). Peak RER were also higher than surgery or non-surgery group (1,20±0,11 versus 1,07±0,13 or 1,08±0,11).

It is expected that these cut-off values would not apply to the present study due to the study population's characteristic differences and a higher proportion of submaximal CPET.

As shown in Table 2, more than half of the patients were below the cut-off values, especially the peak VO2 values.

peak VO2 <18ml/kg/min 75,9% of patients

OUES <1550 57,4% of patients

VE/CO2 slope >31.5 63,1% of patients

This limitation is described in the Discussion: present study were older and most likely weaker than in the study from Coeckelberghs et al.[16] Hence, the cut-offs were probably not appropriate and the predictive value correspondingly underestimated.

So, the present article should focus on new cut-off values and not the previous. Prognostic cut-off values are always linked to population characteristics.

.

The cut-offs with 95% CI derived from the study population for the non-surgery and surgery group were as follow:

peak VO2, 15.7 [11.8 – 18.1] ml/kg/min and 12.5 [9.8 – 15.7];

OUES, 1.75 [1.2 – 2.1] and 1.35 [0.58 –2.26];

VE/CO2-slope, 50.1 [27.4 – 58.6] and 34.2 [31.5 – 38.2].

The cut-off values were different from the previous study and between the study groups (surgery or non-surgery).”

“The 95% CI of our own cut-offs were wide and may therefore not be applicable for other cohorts.”

Agree. A limited submaximal CPET might have influenced it at T0 for some patients.

3) On the abstract, there is information that CPET was performed at the start of CR (T0), end of CR (T1), and 1-year follow-up (T2), but analyses included only T0 and T2 time points.

No results were available for T1. Why?

Analysis should be performed within the 3 CPET groups (T0, T1, and T2).

When treating patients with more severe disease or after procedures, an initial submaximal CPET is expected, so a second CPET (first maximal) is necessary to evaluate these patients better.

Carvalho T, Milani M, Ferraz AS, Silveira AD, Herdy AH, Hossri CAC, et al. Brazilian Cardiovascular Rehabilitation Guideline – 2020. Arq Bras Cardiol. 2020;114(5):943-987.https://doi.org/10.36660/abc.20200407.

Therefore, logistic regression models should include T1 results, besides or instead of T0 results.

4) Table 1 presents an age average of 72.5 (5.3), while the abstract is 73±5.4. Could you please explain its difference?

5) Table 1 has only the baseline characteristics of the overall group. It should also include group characteristics (and possible differences) among surgery and non-surgery groups.

It would be even better if patients’ characteristics were described for each procedure: PCI, No revascularization; Percutaneous valve replacement, surgical valve replacement, and CABG.

Possible doubts and bias:

- Had the surgery group patients more diabetes or lower ejection fraction?

- Had PCI patients more acute coronary syndrome?

- Were percutaneous valve replacement patients older?

- Had percutaneous valve replacement patients more COPD? Or were patients older?

-What exactly is a "No revascularization patient?

Stable CAD? Heart failure? Acute coronary syndrome patient not suitable for revascularization? This is not clear in the article.

6) In the article:

“ Mixed logistic regression models adjusted for age, sex, PCI, time between index event and the start of CR as fixed, centre as random factor and baseline CPET parameters added individually to the model were performed to determine the associations of CPET characteristics with MACE.”

Why was the only PCI included in the regression model?

Why were not other procedures included?

Why was acute coronary syndrome not included?

As described in Table 1: PCI 653 (55%); No revascularization 78 (7%); Percutaneous valve replacement 101 (2%); Surgical valve replacement 79 (7%); CABG 344 (29%).

The manuscript states that "PCI as an indication for CR was associated with MACE (Odds ratio ≈ 1.7)”.

This is because of PCI, or those patients were mainly the ones with the acute coronary syndrome and, consequently, had a higher probability of short-term MACE? This can be a critical bias of the analysis, and this information needs to be more precise.

Possibly there is a requirement to modify and reanalyze the logistic regression models.

7) What was the period between the index event and the CR start?

This data was not described in the article.

CPET completed at a short period after the surgical intervention has limited clinical use and, even more, limited prognostic utility. There can be a risk of complications (bleeding, infection) related to the procedure and not related to baseline diseases.

Esternal pain, exercise discomfort, or even fear to exercise can lead to a submaximal evaluation. The mean peak RER was 1,07 at T0 on the surgery group. So, more than half of the initial CPET were submaximal.

Also, as shown by the actual and previous article, lower hemoglobin levels and limited ventilation response can impact peak VO2 at initial CPET (T0).

That is why I missed the information of CPET at T1, after the end of CR.

A CPET performed 6 to 8 or 12 weeks after the event or procedure could better predict prognosis than an initial limited submaximal CPET in some patients, as previously discussed.

8) What was the duration of CR? There is no information about it.

CR duration affected prognosis. Was it evaluated?

9) MACE reported: 195 patients (14%) within a mean (SD) follow-up time of 340 (112) days.

14 (1%) allcause-mortality,

11 (1%) CV-mortality

1 (0%) aborted sudden cardiac death.

26 (2%) acute coronary syndromes

121 (9%) CV hospitalizations

107 (8%) CV emergency visits

123 (9%) CV interventions.

I missed a visual graphic of MACE versus time on both study groups. (Kaplan-Meier).

Follow-up was short, as written in Discussion. “In our study, the follow-up period may have been too short, and the definition of MACE too wide to obtain a valuable prediction of MACE.”

Agree. Maybe longer follow-up can provide better results.

Hospitalizations, emergency visits, and interventions could be higher in short-term follow-ups of surgical patients and after acute coronary syndrome.

There is a need for more detailed information about MACE reported.

10) “Mean improvement in peak VO2 was 0.25 l/min higher in surgery patients compared to non-surgery patients. However, the difference declined when adjusting for ΔHb, ΔVT1, or ΔHR reserve was more than halved when adjusted for Δpeak TV and disappeared almost completely when adjusted for ΔHb

and Δpeak TV variables (Figure 3).”

Was the Δpeak RER adjust performed?

Peak RER was different between T0 e T2.

A submaximal CPET at T0 might have been compared with a maximal CPET at T2 in the same cases. This could have influenced differences in peak VO2.

15) Using our own cut-offs did not significantly improve the prediction of MACE (AUC = 66.99, specificity = 52.86, sensitivity = 73.84) compared to the established cut-offs (Table 4)

Table 4 was not available in the document for Review.

Conclusions:

It is promising but needs to be clearly described and provide more detailed information, as described previously.

Maybe it is necessary to change the logistic regression models.

A longer follow-up may be necessary.

The definition of MACE was too broad and may include short term procedure complications.

Focus on new cut-off values may be better than using previous values derived from a different population.

6. PLOS authors have the option to publish the peer review history of their article (what does this mean?). If published, this will include your full peer review and any attached files.

Reviewer #1: No

Reviewer #2: No

---

## [Author Response · Author response to Decision Letter 0]

1 Jun 2021

We thank the reviewers for their constructive comments and hope we have addressed them accordingly. Please find below our responses to the reviewers’ comments in the uploaded word document "Response to Reviewers".

---

## [Decision Letter · Decision Letter 1]

19 Jul 2021

Changes and prognostic value of cardiopulmonary exercise testing parameters in elderly patients undergoing cardiac rehabilitation: the EU-CaRE observational study

PONE-D-20-37659R1

Dear Dr. Marcin,

We’re pleased to inform you that your manuscript has been judged scientifically suitable for publication and will be formally accepted for publication once it meets all outstanding technical requirements.

Kind regards,

Gerson Cipriano Jr., PT, MsC, Ph.D.

Academic Editor

PLOS ONE

Additional Editor Comments (optional):

Reviewers' comments:

Reviewer's Responses to Questions

**Comments to the Author**

1. If the authors have adequately addressed your comments raised in a previous round of review and you feel that this manuscript is now acceptable for publication, you may indicate that here to bypass the “Comments to the Author” section, enter your conflict of interest statement in the “Confidential to Editor” section, and submit your "Accept" recommendation.

Reviewer #1: All comments have been addressed

Reviewer #2: All comments have been addressed

2. Is the manuscript technically sound, and do the data support the conclusions?

Reviewer #1: (No Response)

Reviewer #2: Yes

3. Has the statistical analysis been performed appropriately and rigorously? 

Reviewer #1: (No Response)

Reviewer #2: Yes

4. Have the authors made all data underlying the findings in their manuscript fully available?

Reviewer #1: (No Response)

Reviewer #2: Yes

5. Is the manuscript presented in an intelligible fashion and written in standard English?

Reviewer #1: (No Response)

Reviewer #2: Yes

6. Review Comments to the Author

Reviewer #1: (No Response)

Reviewer #2: I enjoyed the new version of the manuscript and several aspects that were previously commented were addressed, and modifications or justifications were made. The research still has several limitations that reduces external validation, but they were reported in the appropriate section. Congratulations for the research and my final recommendation was approval for publication.

7. PLOS authors have the option to publish the peer review history of their article (what does this mean?). If published, this will include your full peer review and any attached files.

Reviewer #1: No

Reviewer #2: No

---

## [Editor Report · Acceptance letter]

26 Jul 2021

PONE-D-20-37659R1 

Changes and prognostic value of cardiopulmonary exercise testing parameters in elderly patients undergoing cardiac rehabilitation: the EU-CaRE observational study 

Dear Dr. Marcin:

I'm pleased to inform you that your manuscript has been deemed suitable for publication in PLOS ONE. Congratulations! Your manuscript is now with our production department. 

Kind regards, 

on behalf of

Professor Gerson Cipriano Jr. 

Academic Editor

PLOS ONE